# Analyses of Early ZIKV Genomes Are Consistent with Viral Spread from Northeast Brazil to the Americas

**DOI:** 10.3390/v15061236

**Published:** 2023-05-25

**Authors:** Laise de Moraes, Moyra M. Portilho, Bram Vrancken, Frederik Van den Broeck, Luciane Amorim Santos, Marina Cucco, Laura B. Tauro, Mariana Kikuti, Monaise M. O. Silva, Gúbio S. Campos, Mitermayer G. Reis, Aldina Barral, Manoel Barral-Netto, Viviane Sampaio Boaventura, Anne-Mieke Vandamme, Kristof Theys, Philippe Lemey, Guilherme S. Ribeiro, Ricardo Khouri

**Affiliations:** 1Programa de Pós-Graduação em Ciências da Saúde, Faculdade de Medicina da Bahia, Universidade Federal da Bahia, Salvador 40026-010, Brazil; laise.moraes@fiocruz.br (L.d.M.);; 2Laboratório de Enfermidades Infecciosas Transmitidas por Vetores, Instituto Gonçalo Moniz, Fundação Oswaldo Cruz, Salvador 40296-710, Brazil; 3Laboratório de Patologia e Biologia Molecular, Instituto Gonçalo Moniz, Fundação Oswaldo Cruz, Salvador 40296-710, Brazil; 4Department of Microbiology, Immunology and Transplantation, Rega Institute for Medical Research, Clinical and Epidemiological Virology, KU Leuven, 3000 Leuven, Belgium; 5Spatial Epidemiology Lab (SpELL), Université Libre de Bruxelles, 1050 Bruxelles, Belgium; 6Department of Biomedical Sciences, Antwerp Institute of Tropical Medicine, 2000 Antwerp, Belgium; 7Escola Bahiana de Medicina e Saúde Pública, Salvador 41150-100, Brazil; 8Instituto de Biología Subtropical, Consejo Nacional de Investigaciones Científicas y Técnicas, Universidad Nacional de Misiones, Puerto Iguazú N3370, Argentina; 9Laboratório de Virologia, Instituto de Ciências da Saúde, Universidade Federal da Bahia, Salvador 40231-300, Brazil; 10Departamento de Patologia e Medicina Legal, Faculdade de Medicina da Bahia, Universidade Federal da Bahia, Salvador 40110-100, Brazil; 11Laboratório de Inflamação e Biomarcadores, Instituto Gonçalo Moniz, Fundação Oswaldo Cruz, Salvador 40296-710, Brazil; 12Hospital Santa Izabel, Salvador 40050-410, Brazil; 13Center for Global Health and Tropical Medicine, Instituto de Higiene e Medicina Tropical, Universidade Nova de Lisboa, 1349-008 Lisbon, Portugal; 14Departamento de Medicina Preventiva e Social, Faculdade de Medicina da Bahia, Universidade Federal da Bahia, Salvador 40110-100, Brazil

**Keywords:** Zika, arboviruses, vector-borne infections, genomic surveillance, phylogenetics

## Abstract

The Americas, particularly Brazil, were greatly impacted by the widespread Zika virus (ZIKV) outbreak in 2015 and 2016. Efforts were made to implement genomic surveillance of ZIKV as part of the public health responses. The accuracy of spatiotemporal reconstructions of the epidemic spread relies on the unbiased sampling of the transmission process. In the early stages of the outbreak, we recruited patients exhibiting clinical symptoms of arbovirus-like infection from Salvador and Campo Formoso, Bahia, in Northeast Brazil. Between May 2015 and June 2016, we identified 21 cases of acute ZIKV infection and subsequently recovered 14 near full-length sequences using the amplicon tiling multiplex approach with nanopore sequencing. We performed a time-calibrated discrete phylogeographic analysis to trace the spread and migration history of the ZIKV. Our phylogenetic analysis supports a consistent relationship between ZIKV migration from Northeast to Southeast Brazil and its subsequent dissemination beyond Brazil. Additionally, our analysis provides insights into the migration of ZIKV from Brazil to Haiti and the role Brazil played in the spread of ZIKV to other countries, such as Singapore, the USA, and the Dominican Republic. The data generated by this study enhances our understanding of ZIKV dynamics and supports the existing knowledge, which can aid in future surveillance efforts against the virus.

## 1. Introduction

Zika virus (ZIKV) is an emerging arthropod-borne virus that can cause a range of symptoms, from common mild arbovirus-like symptoms to severe neurotropic diseases, such as Guillain-Barré and Congenital Zika Syndrome [1]. The virus belongs to the Flavivirus genus of the *Flaviviridae* family. It has a positive-sense single-stranded RNA genome of approximately 11 kb [2,3] and a substitution rate of 7.55 × 10^−4^ to 1.66 × 10^−3^ substitutions per site per year [4]. ZIKV was first isolated in 1947 from a sentinel monkey in the Zika Forest, Uganda. For half a century, ZIKV was described as causing sporadic infections in Africa. In 2007 and 2013, large outbreaks were reported in Micronesia [5] and French Polynesia [6]. In March 2015, ZIKV was identified in Brazil [7,8] and further spread to more than 40 countries in the Americas [9,10]. In February 2016, the World Health Organization (WHO) declared the ZIKV outbreak a Public Health Emergency of International Concern (PHEIC) due to its association with microcephaly and neurological diseases such as Guillain-Barré syndrome [11,12]. This PHEIC declaration was lifted in November 2016 [13].

The revolution in molecular epidemiological surveillance with the advent of next-generation sequencing and advanced phylogenetic analyses allows the uncovering of unobserved transmission dynamics and helps to explain the introduction, spread, and evolution of infectious diseases [11,14]. Molecular dating analyses based on ZIKV sequences from patients identified during the Brazilian outbreak revealed that this virus possibly entered Brazil in the second half of 2013 [14]. Additionally, Lednicky et al. (2016) [15] and Campos et al. (2018) [16] described that ZIKV in Brazil possibly originated from Haiti. Both discoveries revealed, at the time of the event, a gap in our national surveillance system that hampered early detection and interventions to curb the spread of the virus in the country.

However, these findings need validation with new strategies of phylogenetic analyses and new sequences. This is because factors such as the limited number of ZIKV sequences compared to the proportion of cases reported during the outbreak in Brazil and the low sequence quality from the beginning of the epidemic (breadth of coverage <70%) can directly affect the credibility of pathogen transmission dynamics reconstruction [17].

The low number of high-quality sequences available for ZIKV can be explained by several factors, including the low viral load in ZIKV infections [18], low detection of ZIKV-infected patients due to a large number of asymptomatic cases (75% to 80%) or mild/unspecific symptoms [19], the abrupt reduction in ZIKV cases [20], and the challenges faced by the Brazilian health system to perform real-time genomic surveillance. Retrospective sequencing of samples from the early stages of the outbreak in Brazil has the potential to increase the accuracy and precision of transmission dynamics reconstruction, which can help inform public health strategies for future outbreaks.

## 2. Materials and Methods

### 2.1. Study Population

During the ZIKV outbreak, our group actively recruited patients with arboviral-like symptoms in Salvador (n = 948) [21] and Campo Formoso (n = 230) [22], Bahia. A total of 21 cases of viremic ZIKV infection were identified between May 2015 to June 2016 in Bahia, Brazil. The study was conducted in accordance with the Declaration of Helsinki and approved by the Research Ethics Committee of the Faculdade de Medicina/UFBA (CEP/CAAE: 56910516.3.0000.5577; approval number: 1.657.324) and by the Research Ethics Committee of the Instituto Gonçalo Moniz/FIOCRUZ (CEP/CAAE: 55904616.4.0000.0040, approval number 3.363.703; CEP/CAAE: 55904616.4.0000.0040, approval number: 1.642.535).

### 2.2. Nucleic Acid Isolation and RT-qPCR

Whole blood, saliva, and urine samples from 1 to 9 days after the onset of symptoms were sent to Instituto Gonçalo Moniz, FIOCRUZ-BA, Bahia, Brazil, for molecular diagnostics. The viral RNA extraction from plasma, urine, and/or saliva (oral swab was incubated in 200 μL of nuclease-free H_2_O) samples from Campo Formoso was performed with a QIAmp Viral RNA Mini Kit (Qiagen, Germany) following the manufacturer’s recommendations and tested by quantitative RT-PCR (RT-qPCR) using a ZDC Molecular Kit (IBMP, Brazil) or the Trioplex Real-time RT-PCR Assay (CDC, Atlanta, GA, USA). The viral RNA extraction from sera samples from Salvador was extracted using a Maxwell 16 Viral Total Nucleic Acid Purification Kit (Promega, Madison, WI, USA), tested by conventional RT-PCR according to Balm et al. (2012) [23], and the results were confirmed using the Trioplex Real-time RT-PCR Assay (CDC, USA). Samples were selected for sequencing based on a Ct value < 30 and the availability of epidemiological metadata, such as date of sample collection and municipality of residence. A total of 14 samples from Salvador and 7 samples from Campo Formoso, Bahia, were included.

### 2.3. Complete Genome MinION Nanopore Sequencing

A protocol developed by Quick and collaborators [24] was used. The cDNA synthesis was generated using the ProtoScript II First Strand cDNA Synthesis Kit (New England Biolabs, Hitchin, UK) and random hexamer priming. The cDNA generated was subjected to multiplex PCR using Q5 High-Fidelity DNA polymerase (New England Biolabs, UK) and a set of specific primers (ZikaAsian V2) designed by the ZiBRA project (https://github.com/zibraproject/zika-pipeline, accessed on 29 September 2022). Amplicons were purified using 1× AMPure XP beads (Beckman Coulter, High Wycombe, UK) and quantified on a Qubit 3.0 fluorimeter (Life Technologies, Carlsbad, CA, USA) using the Qubit dsDNA BR assay. Library preparation was performed using the Ligation Sequencing Kit SQK-LSK108 (Oxford Nanopore Technologies, Oxford, UK) and Native Barcoding Kit EXP-NBD103 (Oxford Nanopore Technologies, UK). Sequencing libraries were loaded into an R9.4/R9.4.1 flow cell (Oxford Nanopore Technologies, UK), and sequencing data were collected for up to 48 h on a MinION platform (Oxford Nanopore Technologies, UK).

### 2.4. Generation of Consensus Sequences from Nanopore

The fast5 files generated during sequencing were submitted to the pipeline defined by Black and colleagues [25] with minor modifications. In brief, the sequencing data were basecalled on the high-accuracy model performed by Guppy v.3.4.4 (Oxford Nanopore Technologies, UK). The basecalled fastq files with a minimum Q score of 7 were selected for the subsequent demultiplex process using Guppy v.3.4.4 (Oxford Nanopore Technologies, UK). A re-demultiplex process, trimming adapters, and chimeras were performed by Porechop v.0.2.4 (https://github.com/rrwick/Porechop, accessed on 29 September 2022). Furthermore, the assembly was performed by Burrows–Wheeler Aligner (BWA) v.0.7.17-r1188 [26] using NCBI Genbank accession number KJ776791.1 as genome reference. The primer sequences were trimmed with align_trim.py. The assembly was then polished, and the variant calling was performed by nanopolish v.0.11.3 (https://github.com/jts/nanopolish, accessed on 29 September 2022). The consensus sequences were then masked with “N” at regions with coverage depth <20, and the variant candidates were incorporated into the consensus genome using VCFtools v.0.1.16 [27]. The assembly statistics were calculated with SAMtools v.1.10 (using HTSlib 1.10.2) [28] and Seqtk v.1.3-r106 (https://github.com/lh3/seqtk, accessed on 29 September 2022).

### 2.5. Collation of Sequence Dataset

A dataset of ZIKV full or near full-length genomes annotated with the date of sample collection and location was compiled. All ZIKV sequence data were retrieved from GenBank on 24 August 2022, and gbmunge (https://github.com/sdwfrost/gbmunge, accessed on 29 September 2022) was used for extracting FASTA format sequences and the associated metadata. Genotyping was conducted using the Genome Detective Virus Tool [29]. Full-length and near full-length (>7000 pb) ZIKV genomes of the Asian genotype were retained for further analyses. Sequences identified as duplicates were excluded, and only South American isolates were kept for the final dataset. The alignment of the sequence dataset was performed using MAFFT v7.455 [30,31] and manually edited using AliView v.1.26 [32].

### 2.6. Maximum Likelihood Analysis and Temporal Signal Estimation

Maximum Likelihood (ML) phylogenetic analyses were performed using IQ-TREE v.1.6.12 [33] under GTR+F+I+G4 with 1000-replicate ultrafast bootstrap [34]. The best-fitting model was inferred in ModelFinder [35] implemented in the IQ-TREE. The ML trees were visualized and edited using FigTree v.1.4.4 (http://tree.bio.ed.ac.uk/software/figtree, accessed on 29 September 2022). To investigate the evolutionary temporal signal, we applied a regression of root-to-tip genetic distances against the date of sample collection using TempEst v1.5.3 [36], which revealed a sufficient temporal signal (R^2^ = 0.3958) to justify a molecular clock approach. The NCBI GenBank accession numbers for the sequences are listed in Appendix A.

### 2.7. Molecular Clock Phylogenetic Analysis

To explore the evolutionary temporal signal using a molecular clock approach, a discrete phylogeographic analysis was performed using a Bayesian Markov Chain Monte Carlo (MCMC) method as implemented in the BEAST package v1.10.5pre [37,38] with BEAGLE v4.0.0 to improve the computational performance [39]. The substitution process was modeled with a codon-partitioned HKY+G4 model for the coding region of the genome [40,41,42]. An asymmetric discrete trait model integrated with Bayesian Stochastic Search Variable Selection (BSSVS) [43] was specified to reconstruct the spread history, and the number of transitions between locations was estimated with a Markov Jumps approach [39,44,45]. The substitution rate was estimated with an Uncorrelated Lognormal Distribution (UCLD) relaxed-clock model [46]. We employed sampled tip dates to address inexact dates [47]. The Skyride model served as a flexible tree prior [48,49]. We used Tracer v1.7.1 [50] to diagnose mixing and convergence properties. Post-burn-in trees were summarized as a Maximum Clade Credibility (MCC) tree using TreeAnnotator v.1.10.5 pre-implemented in BEAST. Trees were visualized and edited using FigTree v.1.4.4.

## 3. Results and Discussion

The sequencing efforts yielded near full-length sequences (>80% reference coverage, Genbank No. KJ776791.1) for 14 ZIKV-positive samples, with an average sequencing depth ranging from 377× to 1438× (Table 1). The clinical and social-demographic characteristics are listed in Appendix A.

A comprehensive dataset was compiled with 14 new genomes and 206 curated near full-length ZIKV genomes (Appendix A). We performed a time-calibrated discrete phylogeographic reconstruction to explore the geographic spread and transmission of ZIKV in Bahia and Brazil, as shown in Figure 1 and Appendix A.

The newly sequenced ZIKV genomes analyzed in this study are distributed over four distinct clades, containing sequences from diverse regions of Brazil (North, Northeast, and Southeast) and other countries in the world, thus providing valuable insights into the migration patterns of ZIKV.

Ten of the newly sequenced isolates (TRDP238, TRDP256, TRDP257, TRDP274, TRDP282, TRDP300, TRDP309, TRDP317, TRDP333, and ZK0110) clustered in a clade together with genomes from Brazil (Southeast and Northeast), Singapore, and Haiti (Clade I, Figure 1B). The estimated time to the most recent common ancestor (tMRCA) was August 2013, with 95% Bayesian high posterior density (HPD) between January 2013 and February 2014. The Singapore isolate was sequenced from a traveler from São Paulo (Brazil) to Singapore in May 2016. This case was described as the first confirmed instance of ZIKV infection in Singapore [51]. The isolates from Haiti were previously associated with isolates from Brazil by Lednicky et al. (2016) [15] and Campos et al. (2018) [16], which also suggested that the 2014 Haitian ZIKV strain led to the spread of ZIKV in Brazil. However, our phylogeographic reconstruction with the new isolates reveals a paraphyletic clustering of viruses from Haiti with respect to Brazilian lineages, suggesting that ZIKV spread from Brazil to Haiti. In addition, this reconstruction suggests that an outbreak in Salvador was responsible for seeding other locations in Northeast and Southeast Brazil and later spreading to Haiti. Noteworthy, this had already been suggested by Massad et al. (2017) [52], but based only on considering the limited numbers of ZIKV infections in Haiti compared to those obtained from the Brazilian epidemic.

Two of the newly sequenced isolates (TRDP173 and ZK0152) clustered together in a clade with genomes isolated from Brazil (Southeast and North) and the USA (Clade II, Figure 1C), with a tMRCA estimated around February 2014 (95% HPD: July 2013, September 2014). The USA isolate was reported in a study by Grubaugh et al. (2019) [10], which clustered exclusively with Brazilian sequences from the Southeast [10]. However, in our analyses, the USA sequence formed a monophyletic clade with the TRDP173 isolated from Bahia.

The TRDP252 isolate clustered together in a clade with sequences from Brazil (Southeast, North, and Northeast) and the Dominican Republic (Clade III, Figure 1D), with a tMRCA estimated around October 2013 (95% HPD: April 2013, March 2014). This Dominican Republic isolate was reported in a study by Metsky et al. (2017) [53], which clustered with Brazilian sequences, including from the Southeast, similar to our results.

The last isolate (TRDP433), sampled in July 2015, clustered together with sequences from Northeast Brazil, with a tMRCA estimated around March 2013 (95% HPD: September 2013, August 2014). All the isolates in this last clade were from Salvador, with sampling dates between May 2015 and January 2016 (Clade IV, Figure 1E).

By analyzing sequences from other countries with the addition of new sequences from the Northeast region, we provide further evidence supporting the central role of ZIKV migration from the Northeast to the Southeast of Brazil and its subsequent spread outside the country. Unfortunately, the long time between the MRCA and the sequence dates does not allow more confident conclusions about the migration routes of ZIKV.

Although ZIKV was detected almost simultaneously in Brazil, in the Salvador and Natal metropolitan regions, during the period of 2015–2016, before the present study, only five sequences, with >80% reference coverage (Genbank No. KJ776791.1), originating from this region were publicly available. The poor genomic sampling from this crucial moment and the location of the Brazilian ZIKV outbreak limits the power of phylogenetic reconstructions and could bias the interpretation of disease transmission dynamics and may result in incorrect interpretations regarding the origin of ZIKV introduction and its routes of spread in Brazil.

Of note, since 2016, several independent efforts have been made in genomic surveillance of ZIKV to cover most of the Brazilian territories over different time periods, such as the prominent ZiBRA project (Zika in Brazil real-time analysis), which produced more than 140 sequences [24,54,55,56,57,58]. However, these great efforts were unable to rescue enough samples from this specific period due to logistical limitations in storing samples for long periods in Central Public Health Laboratories (LACEN).

## 4. Conclusions

In this study, we provide additional evidence supporting the patterns of ZIKV spread from Northeast Brazil to the Americas by generating 14 additional ZIKV sequences with good coverage (>80%), which represent ~75% of the new publicly available dataset of this period. This augmented dataset allowed us to better understand the dynamics of arbovirus infections worldwide and lends support to future public health responses against this virus. Our conclusions are derived from phylogenetic and phylogeographic analyses, which reveal the relationships among the sequences and the most probable direction of viral spread based on robust statistical analysis. This study contributes to the understanding of the migration history of ZIKV, showing evidence for movement from Northeast into Southeast Brazil and subsequently to other countries such as Haiti, Singapore, the USA, and the Dominican Republic.

## Figures and Tables

**Figure 1 viruses-15-01236-f001:**
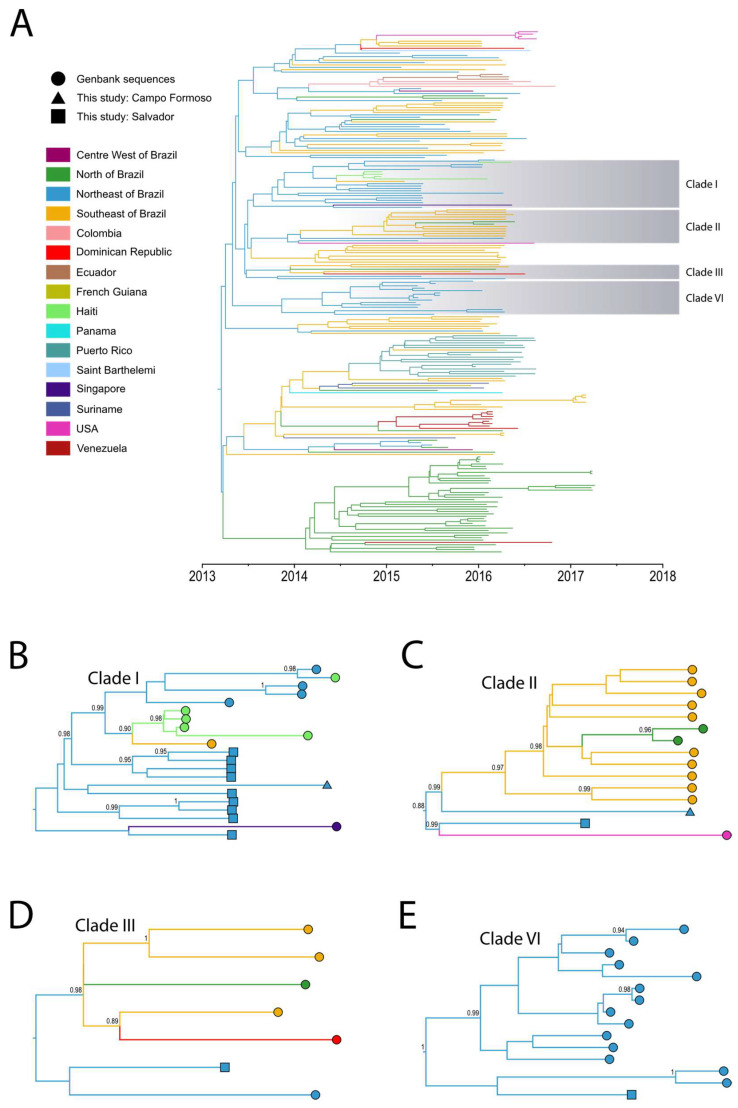
Bayesian phylogeographic reconstruction of ZIKV in Brazil. (**A**) Maximum clade credibility (MCC) phylogeny estimated from full or near full-length ZIKV sequences from Brazil. For visual clarity, clades highlighted in gray represent the sequences reported in this study. Branch colors indicate the most probable ancestral lineage location among countries in the Americas and Brazilian states. (**B**–**E**) Highlighted clades of the inferred geographic migration history of ZIKV, showing the movement from Northeast into Southeast Brazil, and subsequently to other countries such as (**B**) Haiti and Singapore, (**C**) USA, (**D**) the Dominican Republic, and (**E**) a single ZIKV outbreak in Salvador, Northeast Brazil. Circles represent sequences from Genbank; Squares and triangles represent sequences from this study. The numbers show the posterior probabilities >0.80.

**Table 1 viruses-15-01236-t001:** Summary of sample collection, location, Cq value, and sequencing metrics for the 14 ZIKV new genome sequences.

ID	Municipality	Collection Date	Cq Value	No. of Mapped Reads	Avg. Depth Coverage	Reference Covered
TRDP173	Salvador	6 May 2015	30.96	39,617	1207.66	94.03
TRDP238	Salvador	18 May 2015	31.16	44,842	1258.69	90.22
TRDP252	Salvador	19 May 2015	28.64	38,452	1197.17	88.76
TRDP256	Salvador	19 May 2015	29.95	39,530	1170.28	89.10
TRDP257	Salvador	19 May 2015	37.96	58,214	1372.15	83.82
TRDP274	Salvador	20 May 2015	37.84	35,944	1116.41	84.62
TRDP282	Salvador	21 May 2015	35.49	21,026	727.62	88.89
TRDP300	Salvador	22 May 2015	30.03	19,670	678.25	81.89
TRDP309	Salvador	26 May 2015	33.87	20,459	705.72	87.94
TRDP317	Salvador	26 May 2015	33.60	20,414	704.03	87.99
TRDP333	Salvador	27 May 2015	39.07	11,078	377.35	80.62
TRDP433	Salvador	9 July 2015	27.63	60,626	1438.09	96.64
ZK0110	Campo Formoso	8 April 2016	21.91	40,452	1301.48	97.18
ZK0152	Campo Formoso	9 April 2016	34.94	17,820	562.33	84.46

## Data Availability

The new sequences have been deposited in NCBI GenBank under accession numbers OQ727565-OQ727578, and the XML files and datasets analyzed in this study are available in the GitHub repository (https://github.com/khourious/Early-ZIKV-genomes-NE-BR-to-the-Americas, accessed on 13 April 2023).

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
