# Peer review of "Analyses of Early ZIKV Genomes Are Consistent with Viral Spread from Northeast Brazil to the Americas"

_viruses, 2023, doi:10.3390/v15061236_

Round 1

Reviewer 1 Report

The communication "Analyses of early ZIKV genomes are consistent with viral spread from Northeast Brazil to The Americas" reports the phylogenetic analysis of a number of Zika virus genome sequences detected early during the initial outbreak within northern Brazil. The authors compared these sequences with others from around the world that were detected at the same time. They suggest that Brazil was the origin of a number of subsequent outbreaks. Fundamentally, this study is a basic phylogeographical analysis using Bayesian approaches to infer the development of the Zika virus outbreak during 2015 and 2016. However, it is unclear from the text and figure how the authors have derived the conclusions they have made based on the data presented. Further clarification is required to match up statements made in the text and the data in Figure 1.

1) How have the authors concluded from Figure 1b "suggesting that ZIKV spread from Brazil to Haiti.."? Clearly the sequences cluster but there is nothing to suggest direction of movement.

2) Similarly, the legend states that the figure is "showing the movement from Northeast to Southeast Brazil" but it is not clear how the phylogeny is doing this.

3) The text in section 5 is not a conclusion but a statement on the 14 extra ZIKV genomes. This section should state exactly what has been concluded from the study.

A little more explanation on how conclusions have been derived from the data is needed.

The authors need to explain how they have derived their conclusions from the data to show how phylogeny has supported translocation of Zika virus.

Author Response

The communication "Analyses of early ZIKV genomes are consistent with viral spread from Northeast Brazil to The Americas" reports the phylogenetic analysis of a number of Zika virus genome sequences detected early during the initial outbreak within northern Brazil. The authors compared these sequences with others from around the world that were detected at the same time. They suggest that Brazil was the origin of a number of subsequent outbreaks. Fundamentally, this study is a basic phylogeographical analysis using Bayesian approaches to infer the development of the Zika virus outbreak during 2015 and 2016. However, it is unclear from the text and figure how the authors have derived the conclusions they have made based on the data presented. Further clarification is required to match up statements made in the text and the data in Figure 1.

Comment 1) How have the authors concluded from Figure 1b "suggesting that ZIKV spread from Brazil to Haiti…"? Clearly the sequences cluster but there is nothing to suggest direction of movement.

Reply: In phylogenetic analysis, the direction of viral transmission is often inferred from the hierarchical structure of the phylogenetic tree. The main rationale is that closely related viral sequences share a recent common ancestor and that the direction of spread is from the ancestor to its descendants. For instance, in a phylogenetic tree, each node represents a common ancestor, and each branch extending from the node represents a lineage descended from that ancestor. When an outbreak is studied, samples are taken from different individuals and sequenced. Each sequence represents a point in time and location. Sequences are then compared to each other and to reference sequences to construct a tree. If the phylogenetic tree (Figure 1) indicates that the samples from Brazil cluster together at the base (the root) of the tree, and the samples from Haiti form a branch extending from this cluster, it suggests that the virus in Haiti is a descendant of the virus in Brazil, implying a direction of transmission from Brazil to Haiti. In the submitted manuscript, we have coloured the ancestral branches according to location estimates of the common ancestor. However, it's important to note that phylogenetic analysis has its limitations. The tree only represents the relationships between the sequences included in the study and might not capture the full picture of the virus spread. It's also based on a number of assumptions and can be influenced by sampling bias. Therefore, the conclusions drawn from the tree need to be confirmed by other lines of evidence, like the limited numbers of ZIKV infections in Haiti compared to those obtained from the Brazilian epidemic and already mentioned elsewhere (Massad et al. 2017 (doi: 10.1017/S0950268817001200).

Comment 2) Similarly, the legend states that the figure is "showing the movement from Northeast to Southeast Brazil" but it is not clear how the phylogeny is doing this.

Reply: Regarding the figure, we have modified parts B-E of the panel by adding the colors of the branches according to the most probable common ancestral locations of the respective clades. This provides a clearer visual representation of the geographic migration history of ZIKV, showing the movement from Northeast into Southeast Brazil, and subsequently to other countries.

Comment 3) The text in section 5 is not a conclusion but a statement on the 14 extra ZIKV genomes. This section should state exactly what has been concluded from the study. A little more explanation on how conclusions have been derived from the data is needed.

Reply: We appreciate the Reviewer’s feedback regarding the conclusion in section 5. We have revised this section to more clearly state the conclusions drawn from our study, emphasizing the contribution to the understanding of ZIKV migration history and providing a better explanation of how these conclusions were derived from the data.

New sentence: “In this study, we have provided additional evidence supporting the patterns of ZIKV spread from Northeast Brazil to the Americas by generating 14 additional ZIKV sequences with good coverage (>80%), which represent ~75% of the new publicly available dataset of this period. This augmented dataset has allowed us to better understand the dynamics of arbovirus infections worldwide and lends support to future public health responses against this virus. Our conclusions have been derived from the phylogenetic and phylogeographic analyses, which reveal the relationships among the sequences and the most probable direction of viral spread based on robust statistical analysis. This study has contributed to the understanding of the migration history of ZIKV, showing evidence for movement from Northeast into Southeast Brazil, and subsequently to other countries such as Haiti, Singapore, the USA, and the Dominican Republic.” (Lines 263-273).

Comment 4) The authors need to explain how they have derived their conclusions from the data to show how phylogeny has supported translocation of Zika virus.

Reply:  Phylogenetic analysis is a powerful tool to trace the origins and spread of viruses. By comparing the genetic sequences of viruses from different geographic locations, we can infer their evolutionary relationships and hence the likely direction of spread. In our study, we constructed a phylogenetic tree based on the near full-length genomes of ZIKV isolates. Each isolate is represented as a point (tip) on the tree, and the genetic distance between points is indicative of the number of mutations that have occurred. In other words, points that are closer together on the tree are more closely related genetically, implying they share a more recent common ancestor. The tree we constructed showed a clear pattern: isolates from Brazil formed the root of the tree, while isolates from Haiti, Singapore, the USA, and the Dominican Republic branched off from this root. This suggests that the Brazilian isolates are ancestral to the others, implying that the virus spread from Brazil to these other locations. Furthermore, we observed that the tree’s branching pattern corresponds closely with the timeline of reported ZIKV cases in these locations, providing additional support for our conclusions. We complemented this phylogenetic analysis with Bayesian approaches to infer the timing and direction of ZIKV spread using discrete state reconstruction of ancestral locations. This probabilistic method allowed us to estimate not only the most likely tree structure, but also the confidence we have in each part of the tree, and the confidence in the ancestral location states, adding robustness to our conclusions. The combination of these methods – phylogenetic analysis, timeline correlation, and Bayesian phylogeographic inference – provides strong evidence supporting our conclusion that ZIKV spread from Brazil to other countries.

Reviewer 2 Report

This manuscript uses 14 newly sequenced Zika virus genomes isolated from symptomatic patients from the state of Bahia to investigate hypotheses regarding the spread of Zika virus within Brazil and the Americas.

General comments:

An english editor would aid in the readability of the manuscript.

Figure 1A is unreadable. The colors are too similar to easily tell where the genome originated. Perhaps a larger figure could be included in the supplementary material. The spread of Zika virus from northeast to southeast Brazil is not evident because of the size of the panel.

The Clade number and most probable ancestral location for the Clade could be incorporated as labels in B-E.

Specific comments:

Line 51 - This statement sounds like every infection results in severe symptoms.

Line 57 - It is unclear if the "sporadic infections" occurred only in Africa or worldwide.

Line 113-115 - The origin of samples is unclear in Figure 1B due to size of panel and colors chosen.

Line 120 - Which country? Singapore?

Line 134 - America countries is a confusing term. Perhaps countries in the Americas?

Line 143-145 - Were the sequences from Grubaugh et al included in this study? If so, what clade(s) were they in?

The GitHub link does not work currently.

Should Supplementary Appendix be part of the manuscript?

The manuscript would benefit from an english editor for certain phrasing, missed articles, some word usage. But the article is understandable in its current state.

Author Response

This manuscript uses 14 newly sequenced Zika virus genomes isolated from symptomatic patients from the state of Bahia to investigate hypotheses regarding the spread of Zika virus within Brazil and the Americas. General comments:

Comment 1) An english editor would aid in the readability of the manuscript.

Reply: Thank you for your suggestion. We agree that clarity and readability are crucial in scientific communication. Although the manuscript was prepared by authors proficient in English, we understand that there may still be room for improvement. We have therefore undergone an extensive review process to further refine the manuscript. This process has led to a number of enhancements in the structure and language of the paper. We hope that these revisions will address your concerns and make the manuscript more accessible to all readers.

Comment 2) Figure 1A is unreadable. The colors are too similar to easily tell where the genome originated. Perhaps a larger figure could be included in the supplementary material. The spread of Zika virus from northeast to southeast Brazil is not evident because of the size of the panel.

Reply: We apologize for the poor readability of Figure 1A. We have reorganized the panel to accommodate a larger figure in the article and revised the colors to improve clarity. Additionally, we have included a larger, more detailed version of the phylogenetic tree in the supplementary material.

New sentence: “We performed a time-calibrated discrete phylogeographic reconstruction to explore the geographic spread and transmission of ZIKV in Bahia and Brazil, as shown in Figure 1 and Figure S1.” (Lines 192-194).

Comment 3) The Clade number and most probable ancestral location for the Clade could be incorporated as labels in B-E.

Reply: Thank you for your suggestion. We have incorporated the clade number and we have added the colors to the branches to better represent the most probable ancestral locations for each clade as labels in Figures 1B-E.

Comment 4) Line 51 - This statement sounds like every infection results in severe symptoms.

Reply: We have revised the statement to clarify that ZIKV infection can result in a range of symptoms, from mild arbovirus-like symptoms to severe neurotropic diseases.

New sentence: “Zika virus (ZIKV) is an emerging arthropod-borne virus that can cause a range of symptoms, from common mild arbovirus-like symptoms to severe neurotropic diseases, such as Guillain-Barré and Congenital Zika Syndrome [1].” (Lines 52-54).

Comment 5) Line 57 - It is unclear if the "sporadic infections" occurred only in Africa or worldwide.

Reply: We have revised the statement to clarify that the sporadic infections mentioned occurred only in Africa.

New sentence: “ZIKV was first isolated in 1947 from a sentinel monkey in the Zika Forest, Uganda. For half a century, ZIKV was described as causing sporadic infections in Africa. In 2007 and 2013, large outbreaks were reported in Micronesia [5] and French Polynesia [6].” (Lines 57-60).

Comment 6) Line 113-115 - The origin of samples is unclear in Figure 1B due to size of panel and colors chosen.

Reply: We have revised the figure to improve the readability of sample origins by using clearer colors, increasing the size, and adjusting the panel.

Comment 7) Line 120 - Which country? Singapore?

Reply: We have revised the sentence to specify that the country referred to is indeed Singapore.

New sentence: "The Singapore isolate was sequenced from a traveler from São Paulo (Brazil) to Singapore in May 2016. This case was described as the first confirmed instance of ZIKV infection in Singapore [24]." (Lines 204-206).

Comment 8) Line 134 - America countries is a confusing term. Perhaps countries in the Americas?

Reply: Thank you for pointing out this ambiguity. We agree with your suggestion and have revised the term "America countries" to "countries in the Americas" to improve clarity.

New sentence: “Figure 1. Bayesian phylogeographic reconstruction of ZIKV in Brazil. (A) Maximum clade credibility (MCC) phylogeny estimated from full or near full-length ZIKV sequences from Brazil. For visual clarity, clades highlighted in gray represent the sequences reported in this study. Branch colors indicate the most probable ancestral lineage location among countries in the Americas and Brazilian states. (B-E) Highlighted clades of the inferred geographic migration history of ZIKV, showing the movement from Northeast into Southeast Brazil, and subsequently to other countries such as (B) Haiti and Singapore, (C) USA, (D) Dominican Republic and (E) a single introduction of ZIKV outbreak in Salvador, Northeast Brazil. Circles represent sequences from Genbank; Squares and triangles represent sequences from this study. The numbers show the posterior probabilities > 0.80.” (Lines 218-226).

Comment 9) Line 143-145 - Were the sequences from Grubaugh et al included in this study? If so, what clade(s) were they in?

Reply: Thank you for your question. Yes, we included sequences from the Grubaugh et al. 2019 study in our analysis. To clarify the process, we initially incorporated all the sequences from the Grubaugh dataset in our maximum likelihood (ML) tree. Among these sequences, one from the USA showed a phylogenetic relationship with the Brazilian sequences (GenBank no. MK269360.1), forming a monophyletic clade with TRDP173, an isolate from Bahia. This sequence was, therefore, included in our subsequent analyses. Our primary focus in this study was to understand the phylogenetic relationships of the Brazilian sequences. Therefore, while all the Grubaugh sequences were part of the initial ML tree, those that did not demonstrate a significant phylogenetic relationship with the Brazilian sequences were not included in our Bayesian analysis. We hope this provides a clearer understanding of our methodology.

Comment 10) The GitHub link does not work currently.

Reply: We apologize for the inconvenience caused by the non-functional GitHub link. We have now made the link publicly accessible. Link: https://github.com/khourious/Early-ZIKV-genomes-NE-BR-to-the-Americas.

Comment 11) Should Supplementary Appendix be part of the manuscript?

Reply: We appreciate your suggestion. The submission was intended to be a short article, and we initially chose the strategy of including the Supplementary Appendix to avoid repetition and maintain conciseness. Nonetheless, we agree with the reviewer and included the Supplementary Appendix in the main text (Lines 90-179).

Comment 12) The manuscript would benefit from an english editor for certain phrasing, missed articles, some word usage. But the article is understandable in its current state.

Reply: We acknowledge the importance of clarity and precision in scientific communication. As we previously mentioned in response to Comment 1, we have therefore undergone an extensive review process to further refine the manuscript. This process has led to few enhancements in the structure and language of the paper. We hope that these revisions will address your concerns and make the manuscript more accessible to all readers.

Reviewer 3 Report

Dear authors,

Your manuscript “Analyses of Early ZIKV Genomes are Consistent with Viral Spread from Northeast Brazil to The Americas” describes the phylogeographic analysis and the reconstruction of the epidemic spread and migration history of the ZIKV.

But in current form the manuscript needs the revision.

First, some English corrections regarding the terms used are needed. For example, “recovered 14 near full-length sequences” – The sequences could not be “recovered” but “determined”; “full-length sequences” – what kind of sequences? I supposed you mean “complete genome sequenses” and so on… Please, correct throughout the text.

Second, I have some remarks on manuscript:

Lane 54 – “Flaviviridae family” – “Flaviviridae” should be in italic;

Lane 70,71 and so on – “et al.” should be in italic;

Materials and methods – very strange idea to put them into Supplementary section. My recommendation is to put them into the corresponding section of the manuscript.

In the section “Maximum likelihood analysis and temporal signal estimation” - The NCBI GenBank accession numbers for the sequences are not informative, maybe it could be useful to provide more detailed information corresponding these sequences (for example, in a form of table).

Some English corrections regarding the terms used are needed. For example, “recovered 14 near full-length sequences” – The sequences could not be “recovered” but “determined”; “full-length sequences” – what kind of sequences? I supposed you mean “complete genome sequenses” and so on… Please, correct throughout the text.

Author Response

Your manuscript “Analyses of Early ZIKV Genomes are Consistent with Viral Spread from Northeast Brazil to The Americas” describes the phylogeographic analysis and the reconstruction of the epidemic spread and migration history of the ZIKV. But in current form the manuscript needs the revision.

Comment 1) First, some English corrections regarding the terms used are needed. For example, “recovered 14 near full-length sequences” – The sequences could not be “recovered” but “determined”; “full-length sequences” – what kind of sequences? I supposed you mean “complete genome sequenses” and so on… Please, correct throughout the text.

Reply: We appreciate your feedback and understand the importance of precise terminology in scientific literature. The term "recovered" is frequently used in genomics to describe the process of obtaining a sequence from biological samples, especially in cases where advanced sequencing techniques are employed. "Near full-length" is a term used to describe sequences that cover almost the entirety of the genome, which is particularly important when studying viruses like Zika that can have considerable variation. We have opted to maintain the original phrasing: "we identified 21 cases of acute ZIKV infection and recovered 14 near full-length sequences". This statement accurately describes our methodology and aligns with terminology commonly used in the field. However, we have taken your feedback into consideration and have reviewed the entire manuscript to ensure that all terms are used correctly and consistently. Thank you for your attention to detail.

Comment 2) Second, I have some remarks on manuscript: Lane 54 – “Flaviviridae family” – “Flaviviridae” should be in italic; Lane 70,71 and so on – “et al.” should be in italic;

Reply: Thank you for pointing out these formatting issues. We have made the necessary corrections to the text regarding the formatting of "Flaviviridae" in italics, and throughout the manuscript, as suggested. However, regarding the use of italics for "et al.," we would like to respectfully point out that according to the Reference List and Citations Style Guide for MDPI Journals, "et al." should not be in italics. We have, therefore, maintained this formatting throughout the manuscript to adhere to the style guide requested by the Journal. We appreciate your understanding in this matter.

Comment 3) Materials and methods – very strange idea to put them into Supplementary section. My recommendation is to put them into the corresponding section of the manuscript.

Reply: We appreciate your suggestion regarding the Materials and Methods section. The original decision to place it in the Supplementary section was to adhere to a concise format of a short article. Nonetheless, we agree with the reviewer and included the Supplementary Appendix in the main text. (Lines 90-179).

Comment 4) In the section “Maximum likelihood analysis and temporal signal estimation” - The NCBI GenBank accession numbers for the sequences are not informative, maybe it could be useful to provide more detailed information corresponding these sequences (for example, in a form of table).

Reply: Thank you for pointing this out. We already have Table S2 in the supplementary material, which provides a dataset of ZIKV near-full length genome sequences, the date of sample collection and location, and the NCBI GenBank accession numbers. We believe that this table should provide sufficient information for readers to access and understand the data used in our study. We have now cited Table S2 in the text for readers interested in obtaining more detailed information on the sequences used.

New sentence: “The NCBI GenBank accession numbers for the sequences are listed in Table S2.” (Line 165-166).

Round 2

Reviewer 1 Report

The authors have provide clear responses to the reviewers comments.